

# Sharing analysis in the Pawns compiler

Lee Naish

Computing and Information Systems, University of Melbourne, Melbourne, Australia

## ABSTRACT

Pawns is a programming language under development that supports algebraic data types, polymorphism, higher order functions and "pure" declarative programming. It also supports impure imperative features including destructive update of shared data structures via pointers, allowing significantly increased efficiency for some operations. A novelty of Pawns is that all impure "effects" must be made obvious in the source code and they can be safely encapsulated in pure functions in a way that is checked by the compiler. Execution of a pure function can perform destructive updates on data structures that are local to or eventually returned from the function without risking modification of the data structures passed to the function. This paper describes the sharing analysis which allows impurity to be encapsulated. Aspects of the analysis are similar to other published work, but in addition it handles explicit pointers and destructive update, higher order functions including closures and pre- and post-conditions concerning sharing for functions.

## INTRODUCTION

This paper describes the sharing analysis done by the compiler for Pawns (*Naish, 2015*), a programming language that is currently under development. Pawns supports both declarative and imperative styles of programming. It supports algebraic data types, polymorphism, higher order programming and "pure" declarative functions, allowing very high level reasoning about code. It also allows imperative code, where programmers can consider the representation of data types, obtain pointers to the arguments of data constructors and destructively update them. Such code requires the programmer to reason at a much lower level and consider aliasing of pointers and sharing of data structures. Low level "impure" code can be encapsulated within a pure interface and the compiler checks the purity. This requires analysis of pointer aliasing and data structure sharing, to distinguish data structures that are only visible to the low level code (and are therefore safe to update) from data structures that are passed in from the high level code (for which update would violate purity). The main aim of Pawns is to get the benefits of purity for most code but still have the ability to write some key components using an imperative style, which can significantly improve efficiency (for example, a more than twenty-fold increase in the speed of inserting an element into a binary search tree).

There are other functional programming languages, such as ML (*Milner, Tofte & Macqueen, 1997*), Haskell (*Jones et al., 1999*) and Disciple (*Lippmeier, 2009*), that allow destructive update of shared data structures but do not allow this impurity to be

Corresponding author
Lee Naish, lee@unimelb.edu.au

[1] Disciple uses "region" information to augment types, with similar consequences.

encapsulated. In these languages, the ability to update the data structure is connected to its type.[1] For a data structure to be built using destructive update, its type must allow destructive update and any code that uses the data structure can potentially update it as well. This prevents simple declarative analysis of the code and can lead to a proliferation of different versions of a data structure, with different parts being mutable. For example, there are four different versions of lists, since both the list elements and the "spine" may (or may not) be mutable, and sixteen different versions of lists of pairs. There is often an efficiency penalty as well, with destructive update requiring an extra level of indirection in the data structure (an explicit "reference" in the type with most versions of ML and Haskell). Pawns avoids this inefficiency and separates mutability from type information, allowing a data structure to be mutable in some contexts and considered "pure" in others. The main cost from the programmer perspective is the need to include extra annotations and information in the source code. This can also be considered a benefit, as it provides useful documentation and error checking. The main implementation cost is additional analysis done by the compiler, which is the focus of this paper.

The rest of this paper assumes some familiarity with Haskell and is structured as follows. 'An Overview of Pawn' gives a brief overview of the relevant features of Pawns. An early pass of the compiler translates Pawns programs into a simpler "core" language; this is described in 'Core Pawns.' 'The Abstract Domain' describes the abstract domain used for the sharing analysis algorithm, 'The Sharing Analysis Algorithm' defines the algorithm itself and 'Example' gives an extended example. 'Discussion' briefly discusses precision and efficiency issues. 'Related Work' discusses related work and 'Conclusion' concludes.

## AN OVERVIEW OF PAWNS

A more detailed introduction to Pawns is given in (*Naish, 2015*). Pawns has many similarities with other functional languages. It supports algebraic data types with parametric polymorphism, higher order programming and curried function definitions. It uses strict evaluation. In addition, it supports destructive update via "references" (pointers) and has a variety of extra annotations to make impure effects more clear from the source code and allow them to be encapsulated in pure code. Pawns also supports a form of global variables (called state variables) which support encapsulated effects, but we do not discuss them further here as they are handled in essentially the same way as other variables in sharing analysis. Pure code can be thought of in a declarative way, where values can be viewed abstractly, without considering how they are represented. Code that uses destructive update must be viewed at a lower level, considering the representation of values, including sharing. We discuss this lower level view first, then briefly present how impurity can be encapsulated to support the high level view. We use Haskell-like syntax for familiarity.

### The low level view

Values in Pawns are represented as follows. Constants (data constructors with no arguments) are represented using a value in a single word. A data constructor with $N > 0$ arguments is represented using a word that contains a tagged pointer to a block of $N$ words

in main memory containing the arguments. For simple data types such as lists, the tag may be empty. In more complex cases, some bits of the pointer may be used and/or a tag may be stored in a word in main memory along with the arguments. Note that constants and tagged pointers are not always stored in main memory and Pawns variables may correspond to registers that contain the value. Only the arguments of data constructors are guaranteed to be in main memory. An array of size *N* is represented in the same way as a data constructor with *N* arguments, with the size given by the tag. Functions are represented as either a constant (for functions that are known statically) or a closure which is a data constructor with a known function and a number of other arguments.

Pawns has a `Ref t` type constructor, representing a reference/pointer to a value of type `t` (which must be stored in memory). Conceptually, we can think of a corresponding `Ref` data constructor with a single argument, but this is never explicit in Pawns code. Instead, there is an explicit dereference operation: `*vp` denotes the value `vp` points to. There are two ways references can be created: let bindings and pattern bindings. A let binding `*vp = val` allocates a word in main memory, initializes it to `val` and makes `vp` a reference to it (Pawns omits Haskell's `let` and `in` keywords; the scope is the following sequence of statements/expressions). In a pattern binding, if `*vp` is the argument of a data constructor pattern, `vp` is bound to a reference to the corresponding argument of the data constructor if pattern matching succeeds (there is also a primitive that returns a reference to the *i*th element of an array). Note it is not possible to obtain a reference to a Pawns variable: variables do not denote memory locations. However, a variable `vp` of type `Ref t` denotes a reference to a memory location containing a value of type `t` and the memory location can be destructively updated by `*vp := val`.

Consider the following code. Two data types are defined. The code creates a reference to `Nil` (`Nil` is stored in a newly allocated memory word) and a reference to that reference (a pointer to the word containing `Nil` is put in another allocated word). It also creates a list containing constants `Blue` and `Red` (requiring the allocation of two cons cells in memory; the `Nil` is copied). It deconstructs the list to obtain pointers to the head and tail of the list (the two words in the first cons cell) then destructively updates the head of the list to be `Red`.

```
data Colour = Red | Green | Blue
data Colours = Nil | Cons Colour Colours -- like [Colour]

    ...
    *np = Nil                      -- np = ref to (copy of) Nil
    *npp = np                      -- npp = ref to (copy of) np
    cols = Cons Blue (Cons Red *np) -- cols = [Blue, Red]
    case cols of
    (Cons *headp *tailp) ->        -- get ref to head and tail
        *headp :=  Red             -- update head with Red
```

The memory layout after the assignment can be pictured as follows, where boxes represent main memory words and `Ref` and `Cons` followed by an arrow represent pointers (no tag is used in either case):

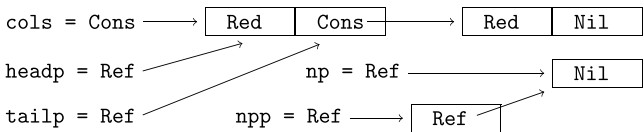

The destructive update above changes the values of both `headp` and `cols` (the representations are shared). One of the novel features of Pawns is that the source code must be annotated with "!" to make it obvious when each "live" variable is updated. If both `headp` and `cols` are used later, the assignment statement above must be written as follows, with `headp` prefixed with "!" and an additional annotation attached to the whole statement indicating `cols` may be updated:

```
*!headp := Red  !cols       -- update *headp (and cols)
```

We say that the statement *directly* updates `headp` and *indirectly* updates `cols`, due to sharing of representations. Similarly, if `headp` was passed to a function that may update it, additional annotations are required. For example, `(assign !headp Red) !cols` makes the direct update of `headp` and indirect update of `cols` clear. Sharing analysis is used to ensure that source code contains all the necessary annotations. One aim of Pawns is that any effects of code should be made clear by the code. Pawns is an acronym for Pointer Assignment With No Surprises.

Pawns functions have extra annotations in type signatures to document which arguments may be updated. For additional documentation, and help in sharing analysis, there are annotations to declare what sharing may exist between arguments when the function is called (a precondition) and what extra sharing may be added by executing the function (called a postcondition, though it is the union of the pre- and post-condition that must be satisfied after a function is executed). For example, we may have:

```
assign :: Ref t -> t -> ()
    sharing assign !p v = _   -- p may be updated
    pre nosharing             -- p&v don't share when called
    post *p = v               -- assign may make *p alias with v
assign !p v =
    *!p := v
```

The "!" annotation on parameter p declares the first argument of `assign` is mutable. The default is that arguments are not mutable. As well as checking for annotations on assignments and function calls, sharing analysis is used to check that all parameters which may be updated are declared mutable in type signatures, and pre- and post-conditions are always satisfied. For example, assuming the previous code which binds `cols`, the call `assign !tailp !cols` annotates all modified variables but violates the precondition

of `assign` because there is sharing between `tailp` and `cols` at the time of the call. Violating this precondition allows cyclic structures to be created, which is important for understanding the code. If the precondition was dropped, the second argument of `assign` would also need to be declared mutable in the type signature and the assignment to p would require v to be annotated. In general, there is an inter-dependence between "!" annotations in the code and pre- and post-conditions. More possible sharing at a call means more "!" annotations are needed, more sharing in (recursive) calls and more sharing when the function returns.

Curried functions and higher order code are supported by attaching sharing and destructive update information to each arrow in a type, though often the information is inferred rather than being given explicitly in the source code. For example, implicit in the declaration for `assign` above is that `assign` called with a single argument of type `Ref t` creates a closure of type `t ->()` containing that argument (and thus sharing the object of type `t`). The explicit sharing information describes applications of this closure to another argument. There is a single argument in this application, referred to with the formal parameter v. The other formal parameter, p, refers to the argument of the closure. In general, a type with $N$ arrows in the "spine" has $K + N$ formal parameters in the description of sharing, with the first $K$ parameters being closure arguments.

The following code defines binary search trees of integers and defines a function that takes a pointer to a tree and inserts an integer into the tree. It uses destructive update, as would normally be done in an imperative language. The declarative alternative must reconstruct all nodes in the path from the root down to the new node. Experiments using our prototype implementation of Pawns indicate that for long paths this destructive update version is as fast as hand-written C code whereas the "pure" version is more than twenty times slower, primarily due to the overhead of memory allocation.

```
data Tree = TNil | Node Tree Int Tree

bst_insert_du :: Int -> Ref Tree -> ()
    sharing bst_insert_du x !tp = _    -- tree gets updated
    pre nosharing                      -- integers are atomic so
    post nosharing                     -- it doesn't share
bst_insert_du x !tp =
    case *tp of
    TNil ->
        *!tp := Node TNil x TNil     -- insert new node
    (Node *lp n *rp) ->
        if x <= n then
            (bst_insert_du x !lp) !tp -- update lp (and tp)
        else
            (bst_insert_du x !rp) !tp -- update rp (and tp)
```

## The high level view

Whenever destructive update is used in Pawns, programmers must be aware of potential sharing of data representations and take a low-level view. In other cases, it is desirable to have a high level view of values, ignoring how they are represented and any sharing that may be present. For example, in the two trees `t1` and `t2` depicted below, it is much simpler if we do not have to care or know about the sharing between the trees and within tree `t1`. The high level view is they are both just `Node (Node TNil 123 TNil) 123 (Node TNil 123 TNil)`.

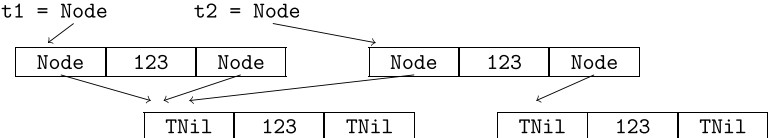

Pawns has a mechanism to indicate that the high level view is taken. Pre- and post-conditions can specify sharing with a special pseudo-variable named `abstract`.[2] The sharing analysis of the Pawns compiler allows a distinction between "abstract" variables, which share with `abstract` and for which the programmer takes a high level view, and "concrete" variables for which the programmer must understand the representation and explicitly declare all sharing in pre- and post-conditions. The analysis checks that no live abstract variables can be destructively updated. Thus, if a function has a parameter which is updated, it must be declared mutable and must not be declared to share with `abstract` in the precondition (non-mutable parameters may or may not share with `abstract`). Checking of preconditions ensures that abstract variables are not passed to functions which expect concrete data structures. For example, an abstract tree cannot be passed to `bst_insert_du` because the precondition allows no sharing with `abstract`. It is important that the tree structure is known when `bst_insert_du` is used because the result depends on it. For example, inserting into the right subtree of `t2` only affects this subtree whereas inserting into the right subtree of `t1` (which has the same high level value) also changes the left subtree of both `t1` and `t2`. Note that concrete variables can be passed to functions which allow abstract arguments. Pawns type signatures that have no annotations concerning destructive update or sharing implicitly indicate no arguments are destructively updated and the arguments and result share with `abstract`. Thus, a subset of Pawns code can look like and be considered as pure functional code.

The following code defines a function that takes a list of integers and returns a binary search tree containing the same integers. Though it uses destructive update internally, this impurity is encapsulated and it can therefore be viewed as a pure function. The list that is passed in as an argument is never updated and the tree returned is abstract so it is never subsequently updated (a concrete tree could be returned if an explicit postcondition without `t = abstract` was given). An initially empty tree is created locally. It is destructively updated by inserting each integer of the list into it (using `list_bst_du`, which calls `bst_insert_du`), then the tree is returned. Within the execution of `list_bst`

[2] There is conceptually a different `abstract` variable for each distinct type.

it is important to understand the low level details of how the tree is represented, but this information is not needed outside the call.

```
data Ints = Nil | Cons Int Ints

list_bst :: Ints -> Tree   -- pure function from Ints to Tree
 -- implicit sharing information:
 -- sharing list_bst xs = t
 -- pre xs = abstract
 -- post t = abstract
list_bst xs =
    *tp = TNil            -- create pointer to empty tree
    list_bst_du xs !tp    -- insert integers into tree
    *tp                   -- return (updated) tree

list_bst_du :: Ints -> Ref Tree -> ()
    sharing list_bst_du xs !tp = _   -- tree gets updated
    pre xs = abstract
    post nosharing
list_bst_du xs !tp =
    case xs of
    (Cons x xs1) ->
        bst_insert_du x !tp  -- insert head of list into tree
        list_bst_du xs1 !tp  -- insert rest of list into tree
    Nil -> ()
```

## CORE PAWNS

An early pass of the Pawns compiler converts all function definitions into a core language by flattening nested expressions, introducing extra variables et cetera. A variable representing the return value of the function is introduced and expressions are converted to bindings for variables. A representation of the core language version of code is annotated with type, liveness and other information prior to sharing analysis. We just describe the core language here. The right side of each function definition is a statement (described using the definition of type `Stat` below), which may contain variables, including function names (`Var`), data constructors (`DCons`) and pairs containing a pattern (`Pat`) and statement for case statements. All variables are distinct except for those in recursive instances of `Stat` and variables are renamed to avoid any ambiguity due to scope.

```
data Stat =                 -- Statement, eg
    Seq Stat Stat |         -- stat1 ; stat2
    EqVar Var Var |         -- v = v1
    EqDeref Var Var |       -- v = *v1
```

```
        DerefEq Var Var |           -- *v = v1
        DC Var DCons [Var] |        -- v = Cons v1 v2
        Case Var [(Pat, Stat)] |    -- case v of pat1 -> stat1 ...
        Error |                     -- (for uncovered cases)
        App Var Var [Var] |         -- v = f v1 v2
        Assign Var Var |            -- *!v := v1
        Instype Var Var             -- v = v1::instance_of_v1_type

data Pat =                          -- patterns for case, eg
        Pat DCons [Var]             -- (Cons *v1 *v2)
```

Patterns in the core language only bind references to arguments — the arguments themselves must be obtained by explicit dereference operations. Pawns supports "default" patterns but for simplicity of presentation here we assume all patterns are covered in core Pawns and we include an error primitive. Similarly, we just give the general case for application of a variable to $N > 0$ arguments; our implementation distinguishes some special cases. Memory is allocated for `DerefEq`, `DC` (for non-constants) and `App` (for unsaturated applications which result in a closure). The runtime behaviour of `Instype` is identical to `EqVar` but it is treated differently in type analysis.

Sharing and type analysis cannot be entirely separated. Destructive update in the presence of polymorphic types can potentially violate type safety or "preservation"— see *Wright (1995)*, for example. For a variable whose type is polymorphic (contains a type variable), we must avoid assigning a value with a less general type. For example, in `*x = []` the type of `*x` is "list of `t`", where `t` is a type variable. Without destructive update, it should be possible to use `*x` wherever a list of any type is expected. However, if `*x` is then assigned a list containing integers (which has a less general type), passing it to a function that expects a list of functions violates type safety ("calling" an arbitrary integer is not safe). Pawns allows expressions to have their inferred types further instantiated using "::", and the type checking pass of the compiler also inserts some type instantiation. The type checking pass ensures that direct update does not involve type instantiation but to improve flexibility, indirect update is checked during the sharing analysis.

## THE ABSTRACT DOMAIN

The representation of the value of a variable includes some set of main memory words (arguments of data constructors). Two variables share if the intersection of their sets of main memory words is not empty. The abstract domain for sharing analysis must maintain a conservative approximation to all sharing, so we can tell if two variables possibly share (or definitely do not share). The abstract domain we use is a set of pairs (representing possibly intersecting sets of main memory locations) of variable *components*. The different components of a variable partition the set of main memory words for the variable.

The components of a variable depend on its type. For non-recursive types other than arrays, each possible data constructor argument is represented separately. For example, the type `Maybe (Maybe (Either Int Int))` can have an argument of an outer `Just`

data constructor, an inner `Just` and `Left` and `Right`. A component can be represented using a list of `x.y` pairs containing a data constructor and an argument number, giving the path from the outermost data constructor to the given argument. For example, the components of the type above can be written as: `[Just.1]`, `[Just.1,Just.1]`, `[Just.1,Just.1,Left.1]` and `[Just.1,Just.1,Right.1]`. If variable v has value `Just Nothing`, the expression `v.[Just.1]` represents the single main memory word containing the occurrence of `Nothing`.

For `Ref t` types we proceed as if there was a `Ref` data constructor, so `vp.[Ref.1]` represents the word vp points to. For function types, values may be closures. A closure that has had $K$ arguments supplied is represented as a data constructor $Cl_K$ with these $K$ arguments; these behave in the same way as other data constructor arguments with respect to sharing, except Pawns provides no way to obtain a pointer to a closure argument. Closures also contain a code pointer and an integer which are not relevant to sharing so they are ignored in the analysis. We also ignore the subscript on the data constructor for sharing analysis because type and sharing analysis only give a lower bound on the number of closure arguments. Our analysis orders closure arguments so that the most recently supplied argument is first (the reverse of the more natural ordering). Consider the code below, where `foo` is a function that is defined with four or more arguments. The sharing analysis proceeds as if the memory layout was as depicted in the diagram. The pre- and post-conditions of `foo` are part of the type information associated with `c1`, `c2` and `c3`.

For arrays, `[Array_.1]` is used to represent all words in the array. The expression, `x.[Array_.1,Just.1]` represents the arguments of all `Just` elements in an array x of `Maybe` values. For recursive types, paths are "folded" (*Bruynooghe, 1986*) so there are a finite number of components. If a type $T$ has sub-component(s) of type $T$ we use the empty path to denote the sub-component(s). In general, we construct a path from the top level and if we come across a sub-component of type $T$ that is in the list of ancestor types (the top level type followed by the types of elements of the path constructed so far) we just use the path to the ancestor to represent the sub-component. Consider the following mutually recursive types that can be used to represent trees which consist of a node containing an integer and a list of sub-trees:

```
data RTrees = Nil | Cons RTree RTrees
data RTree = RNode Int RTrees
```

For type `RTrees` we have the components `[]` (this folded path represents both `[Cons.2]` and `[Cons.1,RNode.2]`, since they are of type `RTrees`), `[Cons.1]` and `[Cons.1,RNode.1]`. The expression `t.[Cons.1,RNode.1]` represents the set of

memory words that are the first argument of `RNode` in variable `t` of type `RTrees`. For type `RTree` we have the components `[]` (for `[RNode.2,Cons.1]`, of type `RTree`), `[RNode.1]` and `[RNode.2]` (which is also the folded version of `[RNode.2,Cons.2]`, of type `RTrees`). In our sharing analysis algorithm we use a function `fc` (fold component) which takes a $v.c$ pair, and returns $v.c'$ where $c'$ is the correctly folded component for the type of variable $v$. For example, `fc (ts.[Cons.2]) = ts.[]`, assuming `ts` has type `RTrees`.

As well as containing pairs of components for distinct variables which may alias, the abstract domain contains "self-alias" pairs for each possible component of a variable which may exist. Consider the following two bindings and the corresponding diagram (as with `Cons`, no tag is used for `RNode`):

```
                          t = RNode ───────▶ ┌──────┬──────┐
                                             │  2   │ Nil  │
t = RNode 2 Nil                              └──────┴──────┘
ts = Cons t Nil                                  ▲
                                                 │
                          ts = Cons ──────▶ ┌──────┬──────┐
                                             │ RNode│ Nil  │
                                             └──────┴──────┘
```

With our domain, the most precise description of sharing after these two bindings is as follows. We represent an alias pair as a set of two variable components. The first five are self-alias pairs and the other two describe the sharing between `t` and `ts`.

```
{{t.[RNode.1], t.[RNode.1]},
 {t.[RNode.2], t.[RNode.2]},
 {ts.[], ts.[]},
 {ts.[Cons.1], ts.[Cons.1]},
 {ts.[Cons.1,RNode.1], ts.[Cons.1,RNode.1]},
 {t.[RNode.1], ts.[Cons.1,RNode.1]},
 {t.[RNode.2], ts.[]}}
```

Note there is no self-alias pair for `t.[]` since there is no strict sub-part of `t` that is an `RTree`. Similarly, there is no alias between `ts.[Cons.1]` and any part of `t`. Although the value `t` is used as the first argument of `Cons` in `ts`, this is not a main memory word that is used to represent the value of `t` (indeed, the value of `t` has no `Cons` cells). The tagged pointer value stored in variable `t` (which may be in a register) is copied into the cons cell. Such descriptions of sharing are an abstraction of computation states. The set above abstracts all computation states in which `t` is a tree with a single node, `ts` is a list of trees, elements of `ts` may be `t` or have `t` as a subtree, and there are no other live variables with non-atomic values.

## THE SHARING ANALYSIS ALGORITHM

We now describe the sharing analysis algorithm. Overall, the compiler attempts to find a proof that for a computation with a depth $D$ of (possibly recursive) function calls, the following condition $C$ holds, assuming $C$ holds for all computations of depth less than $D$. This allows a proof by induction that $C$ holds for all computations that terminate normally.

$C$: For all functions $f$, if the precondition of $f$ is satisfied (abstracts the computation state) whenever $f$ is called, then

1. for all function calls and assignment statements in $f$, any live variable that may be updated at that point in an execution of $f$ is annotated with "!",

2. there is no update of live "abstract" variables when executing $f$,

3. all parameters of $f$ which may be updated when executing $f$ are declared mutable in the type signature of $f$,

4. the union of the pre- and post-conditions of $f$ abstracts the state when $f$ returns plus the values of mutable parameters in all states during the execution of $f$,

5. for all function calls and assignment statements in $f$, any live variable that may be directly updated at that point is updated with a value of the same type or a more general type, and

6. for all function calls and assignment statements in $f$, any live variable that may be indirectly updated at that point only shares with variables of the same type or a more general type.

The algorithm is applied to each function definition in core Pawns to compute an approximation to the sharing before and after each statement (we call it the alias set). This can be used to check points 1, 2, 4 and 6 above. The algorithm checks that preconditions are satisfied for each function call, allowing the induction hypothesis to be used. Point 3 is established using point 1 and a simple syntactic check that any parameter of $f$ that is annotated "!" in the definition is declared mutable in the type signature (parameters are considered live throughout the definition). Point 5 relies on 3 and the type checking pass. The core of the algorithm is to compute the alias set after a statement, given the alias set before the statement. This is applied recursively for compound statements in a form of abstract execution. Note that for point 4, if a statement changes the set of memory cells used to represent a mutable parameter, the algorithm computes the sharing for the union of the two sets of cells.

We do not prove correctness of the algorithm but hope our presentation is sufficiently detailed to have uncovered any bugs. A proof would have a separate case for each kind of statement in the core language, showing that if the initial alias set abstracts the execution state before the statement the resulting alias set abstracts the execution state after the statement. This would require a more formal description of execution states and their relationship with the core language and the abstract domain. The abstract domain relies on type information so the sharing analysis relies on type preservation in the execution. Type preservation also relies on sharing analysis. Thus, a completely formal approach must tackle both problems together. Although our approach is not formal, we do state the key condition $C$, which has points relating to both sharing and types, and we include `Instype` in the core language.

The alias set used at the start of a definition is the precondition of the function. This implicitly includes self-alias pairs for all variable components of the arguments of the function and the pseudo-variables $abstract_T$ for each type $T$ used. Similarly, the postcondition implicitly includes self-alias pairs for all components of the result (and the $abstract_T$ variable if the result is abstract).[3] As abstract execution proceeds, extra

---

[3] Self-aliasing for arguments and results is usually desired. For the rare cases it is not, we may provide a mechanism to override this default in the future.

variables from the function body are added to the alias set and variables that are no longer live can be removed to improve efficiency. For each program point, the computed alias set abstracts the computation state at that point in all concrete executions of the function that satisfy the precondition. For mutable parameters of the function, the sharing computed also includes the sharing from previous program points. The reason for this special treatment is explained when we discuss the analysis of function application. The alias set computed for the end of the definition, with sharing for local variables removed, must be a subset of the union of the pre- and post-condition of the function.

Before sharing analysis, a type checking/inference pass is completed which assigns a type to each variable and function application. This determines the components for each variable. Polymorphism is also eliminated as follows. Suppose we have a function `take n xs`, which returns the list containing the first `n` elements of `xs`:

```
take :: Int -> [a] -> [a]
    sharing take n xs = ys
    pre nosharing
    post ys = xs
```

For each call to `take`, the pre- and post-conditions are determined based on the type of the application. An application to lists of Booleans will have two components for each variable whereas an application to lists of lists of Booleans will have four. When analysing the definition of `take` we instantiate type variables such as `a` above to `Ref ()`. This type has a single component which can be shared to represent possible sharing of arbitrary components of an arbitrary type. Type checking prevents sharing between non-identical types, such as `[a]` and `[b]`. Finally, we assume there is no type which is an infinite chain of refs, for example, `type Refs = Ref Refs` (for which type folding results in an empty component rather than a `[Ref.1]` component; this is not a practical limitation).

Suppose $a_0$ is the alias set just before statement $s$. The following algorithm computes $\text{alias}(s, a_0)$, the alias set just after statement $s$. The algorithm structure follows the recursive definition of statements and we describe it using pseudo-Haskell, interspersed with discussion. The empty list is written `[]`, non-empty lists are written `[a, b, c]` or `a:b:c:[]` and `++` denotes list concatenation. At some points we use high level declarative set comprehensions to describe what is computed and naive implementation may not lead to the best performance.

```
alias (Seq stat1 stat2) a0 =                    -- stat1; stat2
  alias stat2 (alias stat1 a0)
alias (EqVar v1 v2) a0 =                         -- v1 = v2
  let
    self1 = {{v1.c_1,v1.c_2}|{v2.c_1,v2.c_2} ∈ a0}
    share1 = {{v1.c_1,v.c_2}|{v2.c_1,v.c_2} ∈ a0}
  in
     a0 ∪ self1 ∪ share1
```

```
alias (DerefEq v1 v2) a0 =                    -- *v1 = v2
  let
    self1 = {{v1.[Ref.1],v1.[Ref.1]}} ∪
            {{fc(v1.(Ref.1:c₁)),fc(v1.(Ref.1:c₂))}|{v2.c₁,v2.c₂} ∈ a0}
    share1 = {{fc(v1.(Ref.1:c₁)),v.c₂}|{v2.c₁,v.c₂} ∈ a0}
  in
     a0 ∪ self1 ∪ share1
```

Sequencing is handled by function composition. To bind a fresh variable v1 to a variable v2, the self-aliasing of v2 (including aliasing between different components of v2) is duplicated for v1 and the aliasing for each component of v2 (which includes self-aliasing) is duplicated for v1. Binding *v1 to v2 is done in a similar way, but the components of v1 must have Ref.1 prepended to them and the result folded, and the [Ref.1] component of v1 self-aliases. Folding is only needed for the rare case of types with recursion through Ref.

```
alias (Assign v1 v2) a0 =                -- *v1 := v2
  let
    -- al = possible aliases for v1.[Ref.1]
    al = {vₐ.cₐ | {v1.[Ref.1],vₐ.cₐ} ∈ a0}
    -- (live variables in al, which includes v1, must be
    -- annotated with ! and must not share with abstract)
    self1al = {{fc(vₐ.(cₐ++c₁)), fc(v_b.(c_b++c₂))}|
                  vₐ.cₐ ∈ al ∧ v_b.c_b ∈ al ∧ {v2.c₁,v2.c₂} ∈ a0}
    share1al = {{fc(vₐ.(cₐ++c₁)),v.c₂} |
                  vₐ.cₐ ∈ al ∧ {v2.c₁,v.c₂} ∈ a0}
  in if v1 is a mutable parameter then
      a0 ∪ self1al ∪ share1al
    else let
      -- old1 = old aliases for v1, which can be removed
      old1 = {{v1.(Ref.1:d:c₁),v.c₂} | {v1.(Ref.1:d:c₁),v.c₂} ∈ a0}
   in   (a0 \ old1) ∪ self1al ∪ share1al
```

Assignment to an existing variable differs from binding a fresh variable in three ways. First, self-sharing for v1.[Ref.1] is not added since it already exists. Second, v1.[Ref.1] may alias several variable components (the live subset of these variables must be annotated with "!" on the assignment statement; checking such annotations is a primary purpose of the analysis). All these variables end up sharing with v2 and what v2 shares with (via share1al) plus themselves and each other (via self1al). The components must be concatenated and folded appropriately. Third, if v1 is not a mutable parameter the existing sharing with a path strictly longer than [Ref.1] (that is, paths of the form $Ref.1 : d : c_1$) can safely be removed, improving precision. The component v1.[Ref.1] represents the single memory word that is overwritten and whatever the old contents shared with is no longer needed to describe the sharing for v1. For mutable parameters the old value may share with variables from the calling context and we retain this information, as explained

later. Consider the example below, where `t` and `ts` are as before and local variables `v1` and `v3` are references to the element of `ts`. The value assigned, `v2`, is `RNode 3 (Cons (RNode 4 Nil) Nil)`.

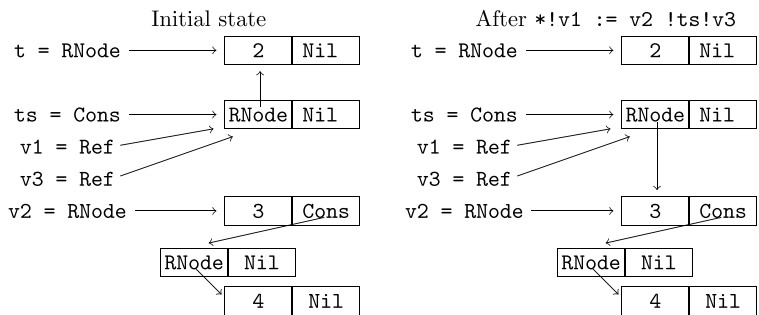

There is aliasing of `v1.[Ref.1]`, `v3.[Ref.1]` and `ts.[Cons.1]` so all these variables have the sharing of `v2` and self-sharing added. Generally we must also add sharing between all pairs of these variables. For example, {`ts.[Cons.1]`, `v3.[Ref.1,RNode.2,Cons.1]`} must be added because the `Cons` component of `v3` did not previously exist. The old sharing of `v1` with `t` is discarded. Note that we cannot discard the old sharing of `ts` and `v3` with `t` for two reasons. First, no *definite* aliasing information is maintained, so we cannot be sure `v3` or `ts` are modified at all. Second, the assignment updates only one memory word whereas there may be other words also represented by `ts.[Cons.1]`. In some cases, the old sharing of `v1` is discarded and immediately added again. Consider the following example, which creates a cyclic list.

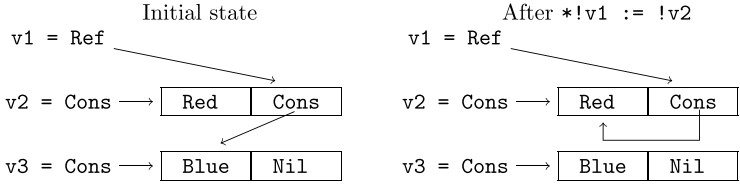

The sharing between `v1` and `v3` is discarded but added again (via `share1al`) because `v2` also shares with `v3`. Correctness of the algorithm when cyclic terms are created depends on the abstract domain we use. A more expressive domain could distinguish between different cons cells in a list. For example, if types are "folded" at the third level of recursion rather than the first, the domain can distinguish three classes of cons cells, where the distance from the first cons cell, modulo three, is zero, one or two. For a cyclic list with a single cons cell, that cons cell must be in all three classes and our algorithm would need modification to achieve this. However, in our domain types are folded at the first level of recursion so we have a unique folded path for each memory cell in cyclic data structure (cyclic terms can only be created with recursive types). There is no distinction between the first and second cons cell in a list, for example.

```
alias (DC v dc [v₁,... vₙ]) a0 =              -- v = Dc v1...vN
  let
    self1 = {{fc(v.[dc.i]), fc(v.[dc.i])} | 1 ≤ i ≤ N} ∪
                {{fc(v.(dc.i:c₁)),fc(v.(dc.j:c₂))}  | {vᵢ.c₁,vⱼ.c₂} ∈ a0}
    share1 = {{fc(v.(dc.i:c₁)),w.c₂} | {vᵢ.c₁,w.c₂} ∈ a0}
  in
      a0 ∪ self1 ∪ share1
```

The `DerefEq` case can be seen as equivalent to $v1 = \texttt{Ref } v2$ and binding a variable to a data constructor with $N$ variable arguments is a generalisation. If there are multiple $v_i$ that share, the corresponding components of v must also share; these pairs are included in `self1`.

```
alias (EqDeref v1 v2) a0 =              -- v1 = *v2
  let
    self1 = {{v1.c₁,v1.c₂} | {fc(v2.(Ref.1:c₁)),fc(v2.(Ref.1:c₂))} ∈ a0}
    share1 = {{v1.c₁,v.c₂} | {fc(v2.(Ref.1:c₁)),v.c₂} ∈ a0
    empty1 = {{v1.[],v.c} | {v1.[],v.c} ∈ (self1 ∪ share1)
  in
    if the type of v1 has a [] component then
      a0 ∪ self1 ∪ share1
    else      --- avoid bogus sharing with empty component
      (a0 ∪ self1 ∪ share1)\ empty1
```

The `EqDeref` case is similar to the inverse of `DerefEq` in that we are removing `Ref.1` rather than prepending it (the definition implicitly uses the inverse of `fc`). However, if the empty component results we must check that such a component exists for the type of `v1`.

```
alias (App v f [v₁,... vₙ]) a0 =              -- v = f v1...vN
  let
    "f(w₁, ... w_{K+ N}) = r" is used to declare sharing for f
    mut = the arguments that are declared mutable
    post = the postcondition of f along with the sharing for
       mutable arguments from the precondition,
       with parameters and result renamed with
       f.[Cl.K],... f.[Cl.1],v₁,... vₙ and v, respectively
    -- (the renamed precondition of f must be a subset of a0,
    -- and mutable arguments of f and live variables they share
    -- with must be annotated with ! and must not share with
    -- abstract)
    -- selfc+sharec needed for possible closure creation
    selfc = {{v.[Cl.i],v.[Cl.i] | 1 ≤ i ≤ N} ∪
                {{v.((Cl.(N+ 1− i)):c₁),v.((Cl.(N+1−j)):c₂)} |
                   {vᵢ.c₁,vⱼ.c₂} ∈ a0} ∪
                {{v.((Cl.(i+ N)):c₁),v.((Cl.(j + N)):c₂)} |
                   {f.((Cl.i):c₁),f.((Cl.j):c₂)} ∈ a0}
    sharec = {{v.((Cl.(N + 1− i)):c₁),x.c₂} | {vᵢ.c₁,x.c₂} ∈ a0} ∪
                {{v.((Cl.(i + N)):c₁),x.c₂ | {f.((Cl.i):c₁),x.c₂} ∈ a0}
    -- postt+postm needed for possible function call
    postt = {{x₁.c₁,x₃.c₃} | {x₁.c₁,x₂.c₂} ∈ post ∧{x₂.c₂,x₃.c₃} ∈ a0}
```

$$\texttt{postm} = \{\{x_1.c_1, x_2.c_2\} \mid \{x_1.c_1, v_i.c_3\} \in \texttt{a0}\} \land \{x_2.c_2, v_j.c_4\} \in \texttt{a0} \land$$
$$\{v_i.c_3, v_j.c_4\} \in \texttt{post} \land v_i \in \texttt{mut} \land v_j \in \texttt{mut}\}$$

```
in
    a0 ∪ selfc ∪ sharec ∪ postt ∪ postm
```

For many `App` occurrences, the function is known statically and we can determine if the function is actually called or a closure is created instead. However, in general we must assume either could happen and add sharing for both. If a closure is created, the first $N$ closure arguments share with the $N$ arguments of the function call and any closure arguments of `f` share with additional closure arguments of the result (this requires renumbering of these arguments).

Analysis of function calls relies on the sharing and mutability information attached to all arrow types. Because Pawns uses the syntax of statements to express pre- and post-conditions, our implementation uses the sharing analysis algorithm to derive an explicit alias set representation (currently this is done recursively, with the level of recursion limited by the fact than pre- and post-conditions must not contain function calls). Here we ignore the details of how the alias set representation is obtained. The compiler also uses the sharing information immediately before an application to check that the precondition is satisfied, all required "!" annotations are present and abstract variables are not modified.

Given that the precondition is satisfied, the execution of a function results in sharing of parameters that is a subset of the union of the declared pre- and post-conditions (we assume the induction hypothesis holds for the sub-computation, which has a smaller depth of recursion). However, any sharing between non-mutable arguments that exists immediately after the call must exist before the call. The analysis algorithm does not add sharing between non-mutable arguments in the precondition as doing so would unnecessarily restrict how "high level" and "low level" code can be mixed. It is important we can pass a variable to a function that allows an abstract argument without the analysis concluding the variable subsequently shares with `abstract`, and therefore cannot be updated. Thus `post` is just the declared postcondition plus the subset of the precondition which involves mutable parameters of the function, renamed appropriately. The last $N$ formal parameters, $w_{K+1}\ldots w_{K+N}$ are renamed as the arguments of the call, $v_1\ldots v_N$ and the formal result $r$ is renamed `v`. The formal parameters $w_1\ldots w_K$ represent closure arguments $K\ldots 1$ of `f`. Thus a variable component such as $w_1$. `[Cons.1]` is renamed `f.[Cl.`$K$`,Cons.1]`.

It is also necessary to include one step of transitivity in the sharing information: if variable components $x_1.c_1$ and $x_2.c_2$ alias in `post` and $x_2.c_2$ and $x_3.c_3$ (may) alias before the function call, we add an alias of $x_1.c_1$ and $x_3.c_3$ (in `postt`). Function parameters are proxies for the argument variables as well as any variable components they may alias and when functions are analysed these aliases are not known. This is why the transitivity step is needed, and why mutable parameters also require special treatment. If before the call, $x_1.c_1$ and $x_2.c_2$ may alias with mutable parameter components $v_i.c_3$ and $v_j.c_4$, respectively, and the two mutable parameter components alias in `post` then $x_1.c_1$ and $x_2.c_2$ may alias

after the call; this is added in `postm`. Consider the example below, where we have a pair `v1` (of references to references to integers) and variables `x` and `y` share with the two elements of `v1`, respectively. When `v1` is passed to function `f1` as a mutable parameter, sharing between `x` and `y` is introduced. The sharing of the mutable parameter in the postcondition, {v1.[Pair.1, Ref.1, Ref.1], v1.[Pair.2, Ref.1, Ref.1]}, results in sharing between `x` and `y` being added in the analysis.

Initial state            After (f1 !v1) !x!y

```
 x = Ref ———————————\              x = Ref ———————————\
v1 = Pair ———————→ Ref Ref         v1 = Pair ———————→ Ref Ref
 y = Ref → Ref        Ref           y = Ref → Ref        Ref
            1    2                            1    2
```

```
f1 :: Pair (Ref (Ref Int)) -> ()
    sharing f1 !v1 = _
    pre nosharing
    post *a = *b; v1 = Pair a b
f1 !v1 =
    case v1 of (Pair rr1 rr2) -> *rr1 := *rr2 !v1
```

The need to be conservative with the sharing of mutable parameters in the analysis of function definitions (the special treatment in `Assign`) is illustrated by the example below. Consider the initial state, with variables `v1` and `v2` which share with `x` and `y`, respectively. After `f2` is called `x` and `y` share, even though the parameters `v1` and `v2` do not share at any point in the execution of `f2`. If mutable parameters were not treated specially in the `Assign` case, `nosharing` would be accepted as the postcondition of `f2` and the analysis of the call to `f2` would then be incorrect. The sharing is introduced between memory cells that were once shared with `v1` and others that were once shared with `v2`. Thus in our algorithm, the sharing of mutable parameters reflects all memory cells that are reachable from the parameters during the execution of the function. Where the mutable parameters are assigned in `f2`, the sharing of the parameters' previous values (`rr1` and `rr2`) is retained. Thus when the final assignment is processed, sharing between the parameters is added and this must be included in the postcondition. Although this assignment does not modify `v1` or `v2`, the "!" annotations are necessary and alert the reader to potential modification of variables that shared with the parameters when the function was called.

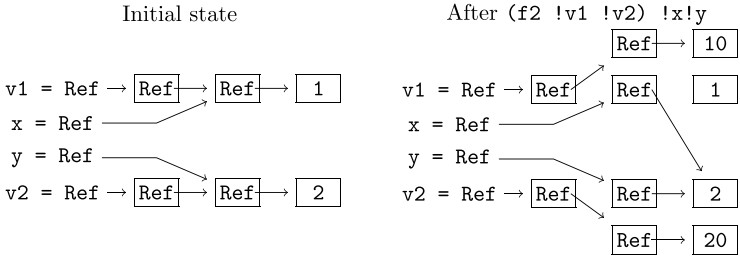

```
f2 :: Ref (Ref (Ref Int)) -> Ref (Ref (Ref Int)) -> ()
    sharing f2 !v1 !v2 = _
    pre nosharing
    post **v1 = **v2
f2 !v1 !v2 =
    *r10 = 10            -- ref to new cell containing 10
    *rr10 = r10          -- ref to above ref
    *r20 = 20            -- ref to new cell containing 20
    *rr20 = r20          -- ref to above ref
    rr1 = *v1            -- save *v1
    rr2 = *v2            -- save *v2
    *!v1 := rr10         -- update *v1 with Ref (Ref 10)
    *!v2 := rr20         -- update *v2 with Ref (Ref 20)
    *rr1 := *rr2 !v1!v2  -- can create sharing at call
```

```
alias Error a0 =  ∅              -- error
alias (Case v [(p₁,s₁),...(pₙ,sₙ)]) a0 =   -- case v of ...
  let
     old = {{v.c₁,v₂.c₂ | {v.c₁,v₂.c₂} ∈ a0}
  in
        ⋃ 1≤ i≤ N aliasCase a0 old v pᵢ sᵢ
```

```
aliasCase a0 av v (Pat dc [v₁,... vₙ]) s = -- (Dc *v1...*vN) -> s
  let
     avdc = {{fc(v.(dc.i:c₁)),w.c₂} | {fc(v.(dc.i:c₁)),w.c₂} ∈ av}
     rself =   {{vᵢ.[Ref.1],vᵢ.[Ref.1]} | 1 ≤ i ≤ N}
     vishare = {{fc(vᵢ.(Ref.1:c₁)),fc(vⱼ.(Ref.1:c₂))} |
                   {fc(v.(dc.i:c₁)),fc(v.(dc.j:c₂))} ∈ av}
     share = {{fc(vᵢ.(Ref.1:c₁)),w.c₂} | {fc(v.(dc.i:c₁)),w.c₂)} ∈ av}
  in
     alias s (rself ∪ vishare ∪ share∪(a0 \ av)∪ avdc)
```

For a case expression we return the union of the alias sets obtained for each of the different branches. For each branch, we only keep sharing information for the variable we are switching on that is compatible with the data constructor in that branch (we remove all the old sharing, `av`, and add the compatible sharing, `avdc`). We implicitly use the inverse of `fc`. To deal with individual data constructors, we consider pairs of components of arguments $i$ and $j$ which may alias in order to compute possible sharing between $v_i$ and $v_j$, including self-aliases when $i = j$. The corresponding component of $v_i$ (prepended with `Ref` and folded) may alias the component of $v_j$. For example, if v of type `RTrees` is matched with `Cons *v1 *v2` and `v.[]` self-aliases, we need to find the components which fold to `v.[]` (`v.[Cons.2]` and `v.[Cons.1,RNode.2]`) in order to compute the sharing for `v2` and `v1`. Thus we compute that `v2.[Ref.1]`, may alias `v1.[Ref.1,RNode.2]`. This can occur if the data structure is cyclic, such as the example below where v is a list containing a single tree with 2 in the node and v as the children (hence it represents a single infinite branch). Note that `v1.[Ref.1,RNode.2]` represents both the memory cell containing the `Cons` pointer and the cell containing `Nil`.

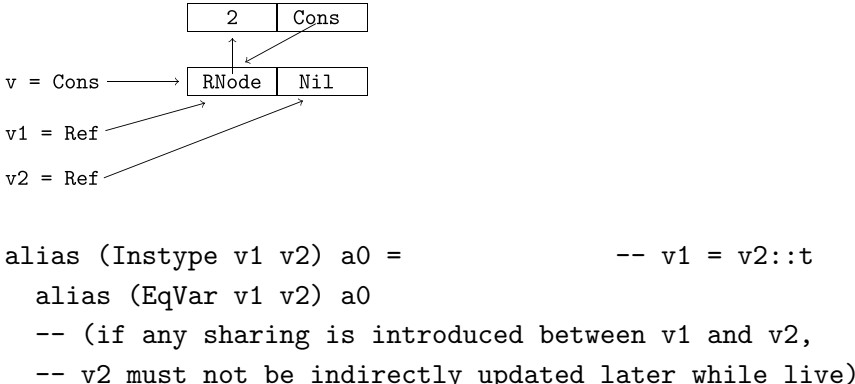

```
alias (Instype v1 v2) a0 =              -- v1 = v2::t
   alias (EqVar v1 v2) a0
   -- (if any sharing is introduced between v1 and v2,
   -- v2 must not be indirectly updated later while live)
```

Type instantiation is dealt with in the same way as variable equality, with the additional check that if any sharing is introduced, the variable with the more general type is not implicitly updated later while still live (it is sufficient to check there is no "!v2" annotation attached to a later statement).

## EXAMPLE

We now show how this sharing analysis algorithm is applied to the binary search tree code given earlier. We give a core Pawns version of each function and the alias set before and after each statement, plus an additional set at the end which is the union of the pre- and post-conditions of the function. To save space, we write the alias set as a set of sets where each inner set represents all sets containing exactly two of its members. Thus $\{\{a, b, c\}\}$ represents a set of six alias pairs: aliasing between all pairs of elements, including self-aliases. The return value is given by variable `ret` and variables `absL` and `absT` are the versions of `abstract` for type `Ints` and `Tree`, respectively.

```
list_bst xs =                          -- 0
    v1 = TNil                          -- 1
    *tp = v1                           -- 2
    list_bst_du xs !tp                 -- 3
    ret = *tp                          -- 4
```

We start with the precondition: $a_0 = \{\{$`xs.[Cons.1]`, `absL.[Cons.1]`$\}$, $\{$`xs.[]`, `absL.[]`$\}\}$. Binding to a constant introduces no sharing so $a_1 = a_0$. $a_2 = a_1 \cup \{$`tp.[Ref.1]`$\}$. The function call has precondition $a_0 \cup \{\{$`tp.[Ref.1]`$\}, \{$`tp.[Ref.1,Node.2]`$\}\}$, which is a superset of $a_2$. Since `tp` is a mutable argument the precondition sharing for `tp` is added: $a_3 = a_2 \cup \{\{$`tp.[Ref.1,Node.2]`$\}\}$. The final sharing includes the return variable, `ret`: $a_4 = a_3 \cup \{\{$`ret.[]`,`tp.[Ref.1]`$\}$, $\{$`ret.[Node.2]`,`tp.[Ref.1,Node.2]`$\}\}$. After removing sharing for the dead (local) variable `tp` we obtain a subset of the union of the pre- and post-conditions, which is $a_0 \cup \{\{$`ret.[]`,`absT.[]`$\}$, $\{$`ret.[Node.2]`, `absT.[Node.2]`$\}\}$.

```
list_bst_du xs !tp =                   -- 0
    case xs of
    (Cons *v1 *v2) ->                  -- 1
```

```
        x = *v1                        -- 2
        xs1 = *v2                      -- 3
        v3 = bst_insert_du x !tp       -- 4
        v4 = list_bst_du xs1 !tp       -- 5
        ret = v4                       -- 6
    Nil ->                             -- 7
        ret = ()                       -- 8
    -- after case                      -- 9
```

We start with the precondition, $a_0 = \{\{\text{tp.}[\text{Ref.}1]\}, \{\text{tp.}[\text{Ref.}1,\text{Node.}2]\},$
$\{\text{xs.}[\text{Cons.}1], \text{absL.}[\text{Cons.}1]\}, \{\text{xs.}[], \text{absL.}[]\}\}$. The Cons branch of the
case introduces sharing for v1 and v2: $a_1 = a_0 \cup \{\{\text{xs.}[\text{Cons.}1], \text{absL.}[\text{Cons.}1],$
$\text{v1.}[\text{Ref.}1], \text{v2.}[\text{Ref.}1,\text{Cons.}1]\}, \{\text{v2.}[\text{Ref.}1], \text{xs.}[], \text{absL.}[]\}\}$. The list
elements are atomic so $a_2 = a_1$. The next binding makes the sharing of xs1 and xs the
same: $a_3 = a_2 \cup \{\{\text{v2.}[\text{Ref.}1], \text{xs.}[], \text{xs1.}[], \text{absL.}[]\}, \{\text{v1.}[\text{Ref.}1], \text{xs.}[\text{Cons.}1],$
$\text{xs1.}[\text{Cons.}1], \text{absL.}[\text{Cons.}1], \text{v2.}[\text{Ref.}1,\text{Cons.}1]\}\}$. This can be simplified by
removing the dead variables v1 and v2. The precondition of the calls are satisfied and
$a_6 = a_5 = a_4 = a_3$. For the Nil branch, we remove the incompatible sharing for xs from
$a_0$: $a_7 = \{\{\text{tp.}[\text{Ref.}1]\}, \{\text{tp.}[\text{Ref.}1,\text{Node.}2]\}, \{\text{absL.}[\text{Cons.}1]\}, \{\text{absL.}[]\}\}$ and
$a_8 = a_7$. Finally, $a_9 = a_6 \cup a_8$. This contains all the sharing for mutable parameter tp and,
ignoring local variables, is a subset of the union of the pre- and post-conditions, $a_0$.

```
bst_insert_du x !tp =                          -- 0
    v1 = *tp                                   -- 1
    case v1 of
    TNil ->                                    -- 2
        v2 = TNil                              -- 3
        v3 = TNil                              -- 4
        v4 = Node v2 x v3                      -- 5
        *!tp := v4                             -- 6
        ret = ()                               -- 7
    (Node *lp *v5 *rp) ->                      -- 8
        n = *v5                                -- 9
        v6 = (x <= n)                          -- 10
        case v6 of
        True ->                                -- 11
            v7 = (bst_insert_du x !lp) !tp     -- 12
            ret = v7                           -- 13
        False ->                               -- 14
            v8 = (bst_insert_du x !rp) !tp     -- 15
            ret = v8                           -- 16
        -- end case                            -- 17
    -- end case                                -- 18
```

Here $a_0 = \{\{\texttt{tp.[Ref.1]}\}, \{\texttt{tp.[Ref.1,Node.2]}\}\}$ and $a_1 = a_0 \cup \{\{\texttt{v1.[]},$ $\texttt{tp.[Ref.1]}\}, \{\texttt{tp.[Ref.1,Node.2]}, \texttt{v1.[Node.2]}\}\}$. For the $\texttt{TNil}$ branch we remove the $\texttt{v1}$ sharing so $a_4 = a_3 = a_2 = a_0$ and $a_5 = a_4 \cup \{\{\texttt{v4.[]}\}, \{\texttt{v4.[Node.2]}\}\}$. After the destructive update, $a_6 = a_5 \cup \{\{\texttt{v4.[]}, \texttt{tp.[Ref.1]}\}, \{\texttt{v4.[Node.2]},$ $\texttt{tp.[Ref.1,Node.2]}\}\}$ ($\texttt{v4}$ is dead and can be removed) and $a_7 = a_6$. For the $\texttt{Node}$ branch we have $a_8 = a_1 \cup \{\{\texttt{v1.[]}, \texttt{tp.[Ref.1]}, \texttt{lp.[Ref.1]}, \texttt{rp.[Ref.1]}\},$ $\{\texttt{tp.[Ref.1,Node.2]}, \texttt{lp.[Ref.1,Node.2]}, \texttt{rp.[Ref.1,Node.2]}, \texttt{v5.[Ref.1]},$ $\texttt{v1.[Node.2]}\}\}$. The same set is retained for $a_9 \ldots a_{17}$ (assuming the dead variable $\texttt{v5}$ is retained), the preconditions of the function calls are satisfied and the required annotations are present. Finally, $a_{18} = a_{17} \cup a_7$, which contains all the sharing for $\texttt{tp}$, and after eliminating local variables we get the postcondition, which is the same as the precondition.

## DISCUSSION

Imprecision in the analysis of mutable parameters could potentially be reduced by allowing the user to declare that only certain parts of a data structure are mutable, as suggested in *Naish (2015)*. It is inevitable we lose some precision with recursion in types, but it seems that some loss of precision could be avoided relatively easily. The use of the empty path to represent sub-components of recursive types results in imprecision when references are created. For example, the analysis of $\texttt{*vp = Nil; v = *vp}$ concludes that the empty component of $\texttt{v}$ may alias with itself and the $\texttt{Ref}$ component of $\texttt{vp}$ (in reality, $\texttt{v}$ has no sharing). Instead of the empty path, a dummy path of length one could be used. Flagging data structures which are known to be acyclic could also improve precision for $\texttt{Case}$. A more aggressive approach would be to unfold the recursion an extra level, at least for some types. This could allow us to express (non-)sharing of separate subtrees and whether data structures are cyclic, at the cost of more variable components, more complex pre- and post-conditions and more complex analysis for $\texttt{Assign}$ and $\texttt{Case}$.

Increasing the number of variable components also decreases efficiency. The algorithmic complexity is affected by the representation of alias sets. Currently we use a naive implementation, using just ordered pairs of variable components as the set elements and a set library which uses an ordered binary tree. The size of the set can be $O(N^2)$, where $N$ is the maximum number of live variable components of the same type at any program point (each such variable component can alias with all the others). In typical code, the number of live variables at any point is not particularly large. If the size of alias sets does become problematic, a more refined set representation could be used, such as the set of sets of pairs representation we used in 'Example,' where sets of components that all alias with each other are optimised. There are also simpler opportunities for efficiency gains, such as avoiding sharing analysis for entirely pure code. We have not stress tested our implementation or run substantial benchmarks as it is intended to be a prototype, but performance has been encouraging. Translating the tree insertion code plus a test harness to C, which includes the sharing analysis, takes less time than compiling the resulting C code using GCC. Total compilation time is less than half that of GHC for equivalent Haskell code and less than one

tenth that of MLton for equivalent ML code. The Pawns executable is around 3–4 times as fast as the others.

## RELATED WORK

Related programming languages are discussed in *Naish (2015)*; here we restrict attention to work related to the sharing analysis algorithm. The most closely related work is that done in the compiler for Mars *Giuca (2014)*, which extends similar work done for Mercury (*Mazur et al., 2001*) and earlier for Prolog *Mulkers (1993)*. All use a similar abstract domain based on the type folding method first proposed in *Bruynooghe (1986)*. Our abstract domain is somewhat more precise due to inclusion of self-aliasing, and we have no sharing for constants. In Mars it is assumed that constants other than numbers can share. Thus, for code such as `xs = []; ys = xs` our analysis concludes there is no sharing between `xs` and `ys` whereas the Mars analysis concludes there may be sharing.

One important distinction is that in Pawns sharing (and mutability) is declared in type signatures of functions so the Pawns compiler just has to check the declarations are consistent, rather than infer all sharing from the code. However, it does have the added complication of destructive update. As well as having to deal with the assignment primitive, it complicates handling of function calls and case statements (the latter due to the potential for cyclic structures). Mars, Mercury and Prolog are essentially declarative languages. Although Mars has assignment statements the semantics is that values are copied rather than destructively updated—the variable being assigned is modified but other variables remain unchanged. Sharing analysis is used in these languages to make the implementation more efficient. For example, the Mars compiler can often emit code to destructively update rather than copy a data structure because sharing analysis reveals no other live variables share it. In Mercury and Prolog, the analysis can reveal when heap-allocated data is no longer used, so the code can reuse or reclaim it directly instead of invoking a garbage collector.

These sharing inference systems use an explicit graph representation of the sharing behaviour of each segment of code. For example, code $s_1$ may cause aliasing between (a component of) variables `a` and `b` (which is represented as an edge between nodes `a` and `b`) and between `c` and `d` and code $s_2$ may cause aliasing between `b` and `c` and between `d` and `e`. To compute the sharing for the sequence $s_1 ; s_2$ they use the "alternating closure" of the sharing for $s_1$ and $s_2$, which constructs paths with edges alternating from $s_1$ and $s_2$, for example `a-b` (from $s_1$), `b-c` (from $s_2$), `c-d` (from $s_1$) and `d-e` (from $s_2$).

The sharing behaviour of functions in Pawns is represented explicitly, by a pre- and post-condition and set of mutable arguments but there is no explicit representation for sharing of statements. The (curried) function `alias s` represents the sharing behaviour of `s` and the sharing behaviour of a sequence of statements is represented by the composition of functions. This representation has the advantage that the function can easily use information about the current sharing, including self-aliases, and remove some if appropriate. For example, in the `[]` branch of the case in the code below the sharing for `xs` is removed and we can conclude the returned value does not share with the argument.

```
map_const_1 :: [t] -> [Int]
    sharing map_const_1 xs = ys pre nosharing post nosharing
map_const_1 xs =
    case xs of
        [] -> xs    -- can look like result shares with xs
        (_:xs1) -> 1:(map_const_1 xs1)
```

There is also substantial work on sharing analysis for logic programming languages using other abstract domains, notably the set-sharing domain of *Jacobs & Langen (1989)* (a set of sets of variables), generally with various enhancements—see *Bagnara, Zaffanella & Hill (2005)* for a good summary and evaluation. Applications include avoiding the "occurs check" in unification (*Søndergaard, 1986*) and exploiting parallelism of independent sub-computations (*Bueno, García de la Banda & Hermenegildo, 1999*). These approaches are aimed at identifying sharing of logic variables rather than sharing of data structures. For example, although the two Prolog goals p(X) and q(X) share X, they are considered independent if X is instantiated to a data structure that is ground (contains no logic variables). Ground data structures in Prolog are read-only and cause no problem for parallelism or the occurs check, whether they are shared or not. The set-sharing domain is often augmented with extra information related to freeness (free means uninstantiated), linearity (linear means there are no repeated occurrences of any variable) and/or groundness (*Bagnara, Zaffanella & Hill, 2005*). In Pawns there are no logic variables but data structures are mutable, hence their sharing is important.

However, the set-sharing domain (with enhancements) has been adapted to analysis of sharing of data structures in object oriented languages such as Java (*Méndez-Lojo & Hermenegildo, 2008*). One important distinction is that Pawns directly supports algebraic data types which allow a "sum of products": there can be a choice of several data constructors (a sum), where each one consists of several values as arguments (a product). In Java and most other imperative and object oriented languages, additional coding is generally required to support such data types. Products are supported by objects containing several values but the only choice (sum) supported directly is whether the object is null or not. Java objects and pointers in most imperative languages are similar to a Maybe algebraic data type, with Nothing corresponding to null. A Ref cannot be null. The abstract domain of *Méndez-Lojo & Hermenegildo (2008)* uses set-sharing plus additional information about what objects are definitely not null. For Pawns code that uses Refs this information is given by the data type—the more expressive types allow us to trivially infer some information that is obscured in other languages. For code that uses Maybe, our domain can express the fact that a variable is definitely Nothing by not having a self-alias of the Just component. The rich structural information in our domain fits particularly well with algebraic data types. There are also other approaches to and uses of alias analysis for imperative languages, such as *Landi & Ryder (1992)* and *Emami, Ghiya & Hendren (1994)*, but these are not aimed at precisely capturing information about dynamically allocated data. A more detailed discussion of such approaches is given in *Giuca (2014)*.

## CONCLUSION

Purely declarative languages have the advantage of avoiding side effects, such as destructive update of function arguments. This makes it easier to combine program components, but some algorithms are hard to code efficiently without flexible use of destructive update. A function can behave in a purely declarative way if destructive update is allowed, but restricted to data structures that are created inside the function. The Pawns language uses this idea to support flexible destructive update encapsulated in a declarative interface. It is designed to make all side effects "obvious" from the source code. Because there can be sharing between the representations of different arguments of a function, local variables and the value returned, sharing analysis is an essential component of the compiler. It is also used to ensure "preservation" of types in computations. Sharing analysis has been used in other languages to improve efficiency and to give some feedback to programmers but we use it to support important features of the programming language.

The algorithm operates on (heap allocated) algebraic data types, including arrays and closures. In common with other sharing analysis used in declarative languages it supports binding of variables, construction and deconstruction (combined with selection or "case") and function/procedure calls. In addition, it supports explicit pointers, destructive update via pointers, creation and application of closures and pre- and post-conditions concerning sharing attached to type signatures of functions. It also uses an abstract domain with additional features to improve precision. Early indications are that the performance is acceptable: compared with other compilers for declarative languages, the prototype Pawns compiler supports encapsulated destructive update, is fast and produces fast executables.

## ACKNOWLEDGEMENTS

Feedback from reviewers, particularly Gianluca Amato, was very helpful in ironing out some important bugs in the algorithm and improving the presentation of this paper.

### Funding

The author declares there was no funding for this work.

### Competing Interests

The author declares there are no competing interests.

### Author Contributions

- Lee Naish conceived and designed the experiments, performed the experiments, analyzed the data, contributed reagents/materials/analysis tools, wrote the paper, prepared figures and/or tables, performed the computation work, reviewed drafts of the paper.

### Data Availability

The following information was supplied regarding the availability of data:

http://people.eng.unimelb.edu.au/lee/src/pawns/.

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
