# Peer review of "Sharing analysis in the Pawns compiler"

_PeerJ Computer Science, doi:10.7717/peerj-cs.22_

## Round 0.1 · original submission · Major Revisions

· Academic Editor

Major Revisions

Please read over the reviews carefully and see whether you can address the supplied comments in a timely manner. There are two aspects, in particular, that your revision should address:
1) Improved clarify of presentation: both reviewers comment on issues regarding clarity, such as the use of notations without prior definition. Please also consider the suggestion to add figures for illustration of the presented concepts.
2) Correctness issues: since the paper does not include a formal proof of correctness, it is very important that the paper provides sufficient evidence to convince the reader. In its current form, the paper raise some correctness concerns as expressed in one of the reviews - please revise the paper in a way that addresses or preempts the raised correctness questions.

Please provide a detailed log of your changes with your revisions. In this log, copy the reviewers' comments and under each comment, state how you have addressed whatever problems or concerns have been pointed out. I am hoping you can do this work expeditiously so that the reviewers can still have your work in mind as they see the revision.

Reviewer 1 ·

Basic reporting

Some notations in the article are not defined:

- In the definition below Line 421 you use the notation ":c" which has not been defined before. Moreover, what is the difference between using square brackets and parentheses, such as "v1.[Ref.1]" and "v1.(Ref.1 :c)"?

- In the definition below Line 426 you use the notation "++c" which has not been defined before. Is it the same as ":c" above? Please define both and explain the difference.

- In the definition below Line 435 you use both the notations "fc(v.[dc.i])" and "fc(v.(dc.i:c))": explain the difference.

Experimental design

No Comments.

Validity of the findings

No Comments.

Additional comments

The paper describes the sharing analysis in the Pawns compiler. The author briefly describes the language and then defines an abstract domain and a sharing algorithm. Since the language contains some extra annotation on the variables which are allowed to share before and after a function call, sharing analysis is used to verify that the annotations cover all the possible sharing.

The author describes the algorithm without a formal proof of correctness (which would help in clarifying some steps of the algorithm but could be mostly case-based and maybe boring).

The results presented in the article are new and correct.

The article is very difficult to read, also because Pawns is a new language. I would strongly suggest to add some figures such as that below Line 132, at least in the example at Line 355 and in the three examples in Section 6, to help the reader.

Moreover, some notations are not defined and should be fixed (see "Basic Reporting" area).

Typos:
- Line 54 mutablity -> mutability
- Line 62: I do not understand the sentence: ".. which source programs are translated into". Please rephrase it.
- Line 443 our the -> our

·

Basic reporting

The article describes the sharing analysis used in the Pawns compiler to validate the correctness of the sharing and mutability annotations. The presentation is mostly self-contained. There are some sentences which are difficult to grasp without having read paper [1] in the bibliography. I have detailed these cases in the "General Comments for the Author" section, but overall they do not affect very much the readability of the paper.

Bibliographic references could be improved. For example, there is a vast literature on the static analysis of sharing for logic programs, but the author only cites the ones which follow the approach based on type folding. A good overview of other approaches to sharing analysis may be found in "Bagnara R, Zaffanella E, Hill PM. Enhanced Sharing Analysis Techniques: A Comprehensive Evaluation. TPLP 5 - 1&2". Some of these analysis have been ported to object oriented languages with destructive updates, such as "Mario Méndez-Lojo, Manuel V. Hermenegildo. Precise set sharing analysis for Java-style programs. Proceedings of VMCAI'08".

Experimental design

No comments.

Validity of the findings

While the general approach followed by the analysis is sound, there is no proof of correctness of the results. This makes very difficult to judge the validity of the work, and increases the possibility that analysis has hidden flaws. I understand that a full correctness proof would be very long and tedious, but some attempts to justify at least the most interesting and tricky constructs (namely, function application and destructive update) should be provided.

In any case, I think that the analysis is mostly correct, but I found some soundness issues which I am going to detail below. The issue with function application is, in my view, the most serious, since it is not clear how to fix it without causing an high loss of precision.

- page 10 line 385-387. I do not think condition 6 as stated here is enough to ensure type safety. Consider the following program

*xsp = []
ysp = xsp::Ref [Int]
zsp = zsp::Ref [Bool]
(assign !ysp [3]) !zsp
a1 = and *zsp

In this case xsp is not alive in line four, hence we do not need to annotate the assign statement with !xsp. The assignment indirectly modifies *zsp to be a list of integers, clearly violating well-typedness. However, condition 6 is respected since [Int] is not a less general type that [Bool]: they are actually incomparable. Therefore, I think that point 6 should be changed in "any live variable that may be indirectly update at that point only shares with variables which have a more general type".

- page 11, alias definition for Assign. Some sharing information is missing on the result, concerning the component v1.[Ref.1]. Consider the following type information:

data Box3 = B3 Int
data Box2 = B2 Box3

and the program fragment:

b3 = B3 0
b3bis = B3 1
b2 = B2 b3
case b2 of
(B2 b3p) ->
*!b3p := b3bis (!b2)

This is the same program annotated with sharing information computed by the analysis. Note that I use the same abbreviations introduced in line 496 of the paper.

b3 = B3 0
< { b3.[B3.1] } >
b3bis = B3 1
< { b3.[B3.1] }, { b3bis.[B3.1] } >
b2 = B2 b3
< { b2.[B2.1] }, { b2.[B2.1,B3.1], b3.[B3.1] }, { b3bis.[B3.1] } >
case b2 of
(B2 b3p) ->
< { b2.[B2.1], b3p.[Ref.1] }, { b2.[B2.1,B3.1], b3.[B3.1], b3p.[Ref.1, B3.1] }, { b3bis.[B3.1] } >
*!b3p := b3bis (!b2)

For the assignment we have:

self1 = < (b3p.[Ref.1], b3p.[Ref.1]), (b3p.[Ref.1,B3.1], b3p.[Ref.1,B3.1]) >
share1 = < (b3p.[Ref.1, B3.1], b3bis.[B3.1]) >
al = { b2.[B2.1] }
selfal = < (b2.[B2.1, B3.1], b2.[B2.1, B3.1]) >
shareal = < (b2.[B2.1, B3.1], b3bis.[B3.1]), (b2.[B2.1, B3.1], b3p.[Ref.1,B3.1]) >
old1 = < (b3p.[Ref.1], b3p.[Ref.1]), (b3p.[Ref.1, B3.1], b3p.[Ref.1, B3.1]), (b2.[B2.1,B3.1], b3p.[Ref.1, B3.1]), (b3.[B3.1], b3p.[Ref.1, B3.1) >
a0 \ old1 = < (b2.[B2.1], b2.[B2.1]), (b2.[B2.1,B3.1], b2.[B2.1, B3.1]), (b3.[B3.1],b3.[B3.1]), (b2.[B2.1,B3.1], b3.[B3.1]), (b3bis.[B3.1], b3bis.[B3.1]) >

result = < { b2.[B2.1] }, { b3p.[Ref.1] }, { b2.[B2.1,B3.1], b3bis.[B3.1], b3p.[Ref.1,B3.1] }, { b2.[B2.1,B3.1], b3.[B3.1] } >

Note that the result does not contain the pair (b2.[B2.1], b3p.[Ref.1]), while these two components share the same location on the heap. In order to fix the definition, it could be enough to add to the result all the pairs between al and v1.[Ref.1], but this requires further investigation.

- page 12, alias definition for DC. I think it is wrong and some pairs are missing from the result, namely:

\bigcup_{i,j \leq 1 \leq N} { {fc(v.(dc.i:c_1)), fc(v.(dc.j:c_2))} | {vi.c1, vj.c2} \in a_0, i != j }

Otherwise, sharing among variables v1, ... vN in not propagated to sharing among the components of v.

- page 13, alias definition for App. I think this is wrong, since posttt does not include all sharing informations which may be created during the execution of f. The problem is that posttt only updates sharing informations on the base of the situation at the end of the function application. However, during the execution, more sharing may be created which is not exposed in the formal arguments at the end.

Consider the following program fragment:

data Box3 = B3 Int
data Box2 = B2 Box3
data Box1 = B1 Box2

bad: Box1 -> Box3 -> ()
sharing bad !b1 b3 = _
pre nosharing
post nosharing

bad !b1 b3 =
case b1 of
(B1 b2p) ->
b2 = *b2p
case b2 of
(B2 b3p) ->
*!b3p := b3 (!b2, !b2p, !b1)
newb3 = B3 0
newb2 = B2 newb3
*!b2p := newb2 (!b1)

Note that the post-condition of the "bad" function is correct, although it cannot be verified by the current analysis algorithm due to loss of precision. Now assume bad is called in the following context.

b3 = B3 0
b3bis = B3 1
b2 = B2 b3
b1 = B1 b2
bad !b1 b3bis

Before the call to "bad", we have the sharing < { b1.[B1.1] }, { b1.[B1.1, B2.1], b2.[B2.1] } , { b1.[B1.1, B2.1, B3.1], b2.[B.2.1, B3.1], b3.[B3.1] }, { b3bis.[B3.1] } >. There is no sharing between b1 and b3bis, hence "bad" may be called. Since the postcondition for bad is "nosharing", the posttt variable does not add any new pair. Therefore, according to the analysis, after we return from "bad" the variables b2 and b3bis do not share. However, this is not true, since inside the execution of "bad" b2 is changed to (B2 b3bis).

It could be possible to argue that this is not a problem because "bad" is not accepted with the given postcondition, but in this way the correctness of the analysis would be dependent of the fact it is not precise enough. This is not a solid ground on which to build the analyzer: any improvement on the analysis could break its correctness. Moreover, I am not sure that it is not possible to provide an example like "bad" which is accepted by the analysis.

This might be fixed by either changing the definition of "alias" for the function application case, or by changing the meaning of the post-condition, so that it records not only the last sharing situation but also (at some extent which should be investigated) the intermediate situations during the execution of the function.

Additional comments

==> Presentation

- page 2 line 50-51. In the sentence "may lead to a proliferation of different versions of data structures" it is not clear what are the different versions you are talking about. I think you are referring to the situation described in page 8 of [*], where different versions of the same function only differ for the set of mutable arguments. This needs to be stated more explicitly.

- page 2 line 52. It is not clear what "destructive update requiring an extra level of indirection" means. I think you are referring to the extra level of indirection you get when you replace every constructor argument with references, as discussed in page 9 of [*]. However, you should be more explicit, because in most languages you may destructively update a non-reference type, and therefore you do not need references everywhere.

- page 6. Somewhere in this page you should state that for those variables which share with abstract you do not keep or track further sharing information. Moreover, if I am not wrong, when you call a function f and an actual parameter is abstract, the corresponding formal parameter should be abstract too. I see this is a consequence of respecting the preconditions for f, however explicitly recognizing this property helps understanding what abstract really means.

- page 8 lines 285-294. Not everyone is familiar with polymorphic types: I would devote more space to explain the problem with destructive updates of polymorphic types. The discussion and the examples in Section 10 of [*] are quite good for this purpose.

- page 10 line 376-387. I would add another point to the list: "for all function calls, the sharing information among the actual parameters is a subset of the sharing information among formal parameters as declared in the pre-condition, modulo variable renaming". The fact that checking pre-conditions is needed is acknowledged later, in line 391, but I do not see any reason why not put it in the list.

- page 11 lines 408-410. There are several things I do not understand here. First of all, what do you mean by "type assignment"? I think it is the correspondence between variables and types computed by the type-checking phase, but I am not sure. Furthermore, you say that "sharing information is given for all type instances of all defined functions". But sharing information should be computed by the analyzer, so what is this sharing information which is given a priori? Are you referring to sharing information in the pre- and post-conditions? Or to sharing information attached to closure types? Moreover, it seems that for polymorphic functions you have many sharing informations for different instances of the type variables. How is it possible?

- page 11 line 410. Here it seems you allow to declare sharing among parameters of different type variables, while in [*] this is explicitly disallowed. A comment would be welcome.

- page 11, alias definition for DerefEq. You introduce the notation (Ref.1 :c) without defining it.

- page 12, alias definition for Assign. You introduce the notation c_a ++ c without defining it.

- page 12, alias definition for Assign. Please, explain why we need to keep old1 when the assignment might create a cyclic structure, since I was not able to come up with an example where it was really needed.

- page 13, alias definition for App, definitions of sharec and self. In order to be consistent with the reverse ordering of closure arguments, I think it is better to replace the components v.[Cl.i] and v.(Cl.i :c) with v[Cl.N-i+1] and v(Cl.N-i+1 :c) respectively.

- page 13 lines 453-454. I think to understand the role of closure arguments, but I cannot grasp the meaning of this sentence.

- page 14 lines 458-461. This sentence puzzled me until I realized that abstract variables and non-mutable variables are two different things. Probably you should stress out this fact somewhere.

- page 14 aliasCase definition. To be consistent (merely from a stylistic point of view) with the alias function for DC, the definitions for avdc, vishare and share should be preceded by \bigcup_{1 \leq i \leq N}. (and also \bigcup_{1 \leq j \leq N} for vishare). Alternatively, the union may be removed in the alias definition of DC.

==> Typos:

- page 7 line 244. The entire line should be removed.

- page 12, alias definition for DC. In the first line of self1, dc is in italic font but it should be in typewriter.

- page 13, alias definition for App. The definitions for selfc and sharec are full of spurious ")"

- page 13 line 443. "our the implementation": either "our" or "the" should be removed.

- page 13 line 454: prepresentation => representation

- page 14 line 461: transisitivity => transitivity

- page 14 aliasCase definition. In the definition of vishare and share the indexes i and j are in typewriter font while they should be in italic.

---

## Round 0.2 · Minor Revisions

· Academic Editor

Minor Revisions

The reviewers found this revision substantially improved and were pleased with how their previous feedback was incorporated in this revision. We can now recommend acceptance. However, one of the reviewer points out a number of further possible improvements. The paper doesn't need another review but I would like to request to address these final comments in the preparation of the final version, just to make this paper even stronger.

Reviewer 1 ·

Basic reporting

No Comments

Experimental design

No Comments

Validity of the findings

No Comments

Additional comments

I have very appreciated the effort of the author to improve readability of the paper with the added examples and explanations. Also the use of "aliasing" instead of "sharing" seems more appropriate. I think that the article can be accepted without further revisions.

Minor points:

Line 478: the definition of the function "take" should be in a separate line
Line 502: write explicitly that in old1 (in Assign) :d:c_1 means that the length is at least one.
Line 547: in selfc (in App) remove the extra parenthesis after v_i.c_1,v_j.c_2
Line 832: I disagree that set-sharing is usually augmented with groundness. Even if [11] provides some evidence that groundness information is useful, often sharing is augmented with linearity or freeness information ([11] is a work of 10 years ago, there has been much work on this topic, such as finding improved algorithms for exploiting linearity and/or freeness in sharing).
Line 840: This claim is too strong, I think that Java has sum through subclasses.

·

Basic reporting

The paper has been vastly improved w.r.t the previous version: not only it is much more readable, but also many bugs have been fixed. I also like the fact that the term "sharing pair" has been replaced with "aliasing pair": I find the latter much more precise. However, I think there are still some problems from the point of view of correctness which deserve a minor revision.

Experimental design

No comments.

Validity of the findings

--> Algorithm for "assign"

There is a correctness issue I overlooked the first time I reviewed the paper. If "v2.c1" and "v2.c2" may share, after *v1:=v2 the same should hold for "v1.(Ref 1 : c1)" and "v1.(Ref.1 : c2)". However, this does not happen with the current analysis algorithm. It should be easy to fix.

--> General correctness of the analysis

Apart from the bug described above, I think the analysis algorithm is correct. However, condition C on page 11, which is the key inductive hypotheses for a possible correctness proof, is not complete. The problem lies at point 4 of C: "the union of pre- and post-conditions of f abstracts the state when f returns plus the values of mutable parameters in all states during the execution of f". This condition is not strong enough to ensure correctness of the algorithm for App, and the evidence is the program f2 on page 19. During the execution of this program, v1 and v2 never share. Therefore, if we only require that the post-condition abstracts the values of the parameters in all the states during the execution, a valid post-condition for f2 would be "nosharing". But this would make the App algorithm incorrect.

Nonetheless, the analysis works because "nosharing" is not accepted as the post-condition of f2, and the reason is that the invariant which is maintained by the analysis is more complex than what is stated at point 4. In lines 582-584 the paper says that parameters are actually "proxies for the argument variables as well as any variable components they may alias". This fact is essential to prove correctness, and should be part, in some way or another, of the condition C. One possibility could be to add something like this to C: "If v1 and v2 are mutable parameters, before the execution va.ca is an alias for v1.c1 and vb.cb is an alias for v2.c2, and the execution of f may make va.ca and vb.cb aliases, then (v1.c1, v2.c2) is in the union of pre- and post-condition". This is enough to prove correctness of App, and I think is preserved by all the other operations (although I suspect Assign is probably too conservative, but this is not a correctness problem).

Additional comments

First of all, I am sorry that my previous review did not contain the bibliography data for [*]: it was a reference to "An informal introduction to Pawns: a declarative/imperative language", which you have cited as reference [1].

--> Expressing sharing properties

It is not always easy to express pre- and post-condition using Pawns' syntax. For example, assume you want to define the function zip with the following pre- and post-conditions:

zip:: [a] -> [a] -> [Pair a a]
sharing zip xs ys = zs
pre ... elements of xs and ys may share, but the "spine" cannot ...
post ... first and second elements of each pair in zs may share ....

Is it possible to express this properties concisely? For the pre-condition, something like this could work:

Case xs of
Cons(*x, *xs1) ->
Case ys of ->
Cons(*y, *ys1) -> *x = *y; ()
Nil -> ()
Nil -> ()

But it is not compact at all (and maybe is also incorrect). I think you should add (not necessarily for this paper) the possibility of expressing sharing as a set of pairs, i.e., with the same representation used by the alias algorithm.

--> Minor points

- page 7 line 252. This sentence is confusing. A concrete tree is returned for any post-condition different than abstract, not just for nosharing.

- page 8 line 313. I would move the first sentence of this paragraph either at the end of the paragraph (replacing InsType with ::), or at the end of the previous paragraph (where InsType is defined).

- page 11 point 5. You have added this point, following my suggestion. Actually, now that I have read the answer in your rebuttal letter, I see the reason why this point was not there in the first place. So, I would not object to the removal of point 5 (but I think it is fine to keep it, if you want).

- page 12 line 459. Why you feel the need to explicitly state that Instype is in the core language?

- page 12 lines 478-479. Please, reformat the "take" function on separate lines, as for all the other examples.

- page 12 lines 480-483. It is still not clear how pre- and post-conditions for the application are generated by pre- and post-conditions in the definition of the function. For example, assume that v has type [ Pair (Ref Int) (Ref Int) ] ( a list of pairs of references to int ). Assume you know that v.[Cons.1, Pair.1, Ref.1] and v.[Cons.1, Pair.2, Ref.1] do not share. Is it possible to infer that, after "y = take 5 v", also y.[Cons.1, Pair.1, Ref.1] and y.[Cons.1, Pair.2, Ref.1] do not share. I would say yes, because since "take" is polymorphic, it cannot "open" or "generate" new pairs. Is it true or there are corner cases I am overlooking?

- page 13 lines 484-486. This is related to the question about line 410 on my previous review. In the answer to that question, you say that type checking prevents sharing between things whose types are distinct variables. I think this means that the following declaration would not type check since xs=ys is not allowed in the pre-condition section.

zip:: [a] -> [b] -> [Pair a b]
sharing zip xs ys = zs
pre xs = ys
post ...whatever works...

However, the sentence "can be shared to represent possible sharing of arbitrary components of an arbitrary type" suggests the opposite. I think you should clearly state in the paper what you wrote in the rebuttal letter, i.e., that type checking prevents sharing between things whose types are distinct variables.

- page 14, algorithm for Assign. It should be possible to simplify the algorithm by removing self1 and share1. Since Ref's cannot be null, { v1.[Ref.1], v1.[Ref.1] } is always in a0, hence almost all the aliasing pairs in self1 and share1 also appear in selfal and shareal. The only exception is { v1.[Ref.1], v1.[Ref.1] } which is in self1 but might not be in selfal. However, { v1.[Ref.1], v1.[Ref.1] } is in a0 \setminus old1, hence it also appears in the result.

- page 14, algorithm for Assign. Please, make it clear that ":d :c1" in the definition of "old1" is not a typo, but a way to ensure that all the components in old1 have at least one element after Ref.1.

- page 14, line 515. I would start another paragraph just before "Consider the example below".

- page 15, line 535. Is "modula" correct in English ?

- page 16, algorithm for App. There is a spurious ")" in the third line of the definition for "selfc".

- page 17 line 553. I would start a new paragraph just before "Analysis of function calls". As an alternative, a preposition such as "On the contrary" or "Instead" should separate this part from the previous one devoted to the case when application generates a closure.

- page 23, end of section 7. You are making comparisons among different compilation and execution times, but you do not give any concrete value. Are the times involved big enough that the differences are significant (in the sense, for example, that results are not biased by the start-up delay of different compilers and runtime systems) ?

- page 24 lines 835-854. I do not fully agree with your statement that Java and other object oriented languages do not have "sum of products". Subtyping through inheritance is not very dissimilar. It may be used to simulate abstract data types, although standard OO languages do not have convenient syntax constructs to work with them (but others, such as Scala, do). In any case, my point is that your abstract domain, with the appropriate changes, could be useful even with standard OO languages.

---

## Round 0.3 · accepted · Accept

· Academic Editor

Accept

Thank you for providing a revision that addresses the final reviewer's comments. We are pleased to see the improved submission and recommend acceptance.